# Long-Term Evolution of Malnutrition and Loss of Muscle Strength after COVID-19: A Major and Neglected Component of Long COVID-19

**DOI:** 10.3390/nu13113964

**Published:** 2021-11-06

**Authors:** Marine Gérard, Meliha Mahmutovic, Aurélie Malgras, Niasha Michot, Nicolas Scheyer, Roland Jaussaud, Phi-Linh Nguyen-Thi, Didier Quilliot

**Affiliations:** 1Transversal Nutrition Unit, University of Lorraine, Nancy University Hospital, 54500 Vandoeuvre-les-Nancy, France; gerard_marine@hotmail.fr (M.G.); m.mahmutovic@chru-nancy.fr (M.M.); a.malgras@chru-nancy.fr (A.M.); n.michot@chru-nancy.fr (N.M.); n.scheyer@chru-nancy.fr (N.S.); 2Internal Medicine and Clinical Immunology Department, University of Lorraine, Nancy University Hospital, 54500 Vandoeuvre-les-Nancy, France; R.Jaussaud@chru-nancy.fr; 3Medical Evaluation Department, Department of Clinical Research Support PARC, University of Lorraine, Nancy University Hospital, 54500 Vandoeuvre-les-Nancy, France; pl.nguyen-thi@chru-nancy.fr

**Keywords:** long COVID-19, muscle strength, malnutrition, self-evaluation, obesity, cohort study, performance status, intensive care unit

## Abstract

Post-acute consequences of COVID-19, also termed long COVID, include signs and symptoms persisting for more than 12 weeks with prolonged multisystem involvement; most often, however, malnutrition is ignored. Method: The objective was to analyze persistent symptoms, nutritional status, the evolution of muscle strength and performance status (PS) at 6 months post-discharge in a cohort of COVID-19 survivors. Results: Of 549 consecutive patients hospitalized for COVID-19 between 1 March and 29 April 2020, 23.7% died and 288 patients were at home at D30 post-discharge. At this date, 136 of them (47.2%) presented persistent malnutrition, a significant decrease in muscle strength or a PS ≥ 2. These patients received dietary counseling, nutritional supplementation, adapted physical activity guidance or physiotherapy assistance, or were admitted to post-care facilities. At 6 months post-discharge, 91.0% of the 136 patients (n = 119) were evaluated and 36.0% had persistent malnutrition, 14.3% complained of a significant decrease in muscle strength and 14.9% had a performance status > 2. Obesity was more frequent in patients with impairment than in those without (52.8% vs. 31.0%; *p* = 0.0071), with these patients being admitted more frequently to ICUs (50.9% vs. 31.3%; *p* = 0.010). Among those with persistent symptoms, 10% had psychiatric co-morbidities (mood disorders, anxiety, or post-traumatic stress syndrome), 7.6% had prolonged pneumological symptoms and 4.2% had neurological symptoms. Conclusions: Obese subjects as well as patients who have stayed in intensive care have a higher risk of functional loss or undernutrition 6 months after a severe COVID infection. Malnutrition and loss of muscle strength should be considered in the clinical assessment of these patients.

## 1. Introduction

Post-acute consequences of COVID-19, also termed long COVID, include signs and symptoms that develop during or after an infection consistent with COVID-19, and persist for more than 12 weeks with prolonged multisystem involvement and significant disability. The most commonly described symptoms are sensory (loss of taste and anosmia), neurological (loss of concentration and “brain fog”) and cardiorespiratory problems (fatigue, dyspnea, reduced exercise capacity). In most instances, malnutrition is left ignored as these patients are often overweight at the time of diagnosis [1].

Recent studies have reported a high prevalence of malnutrition among hospitalized patients with COVID-19 depending on the screening and diagnostic tool used [2], and levels are estimated at about 50% (31.7–66.5%) [2]. This prevalence furthermore appears particularly high in patients requiring a stay in intensive care [1,3]. Several factors could explain malnutrition during this acute phase, including marked systemic inflammation driving hypermetabolism and muscle catabolism, and prolonged periods of bedrest driving disuse atrophy. Up to 40% of patients with COVID-19 experience gastrointestinal symptoms ranging from nausea, vomiting, anorexia, diarrhea and abdominal distention, especially in ICU COVID patients [4], which can further deter eating and impact the tolerance of nutritional support [5]. Olfactory and gustatory dysfunction [6] may also contribute to weight loss [7,8].

Muscle loss and/or loss of muscle function appear to be the major nutritional challenges during this acute period. Both myalgia and muscle loss have been strongly correlated with disease severity among COVID-19 patients [9]. Protein turnover is also increased in critical illness in the early stages of COVID-19, in response to massive proteolytic stimuli [10]. Suggested mechanisms include direct muscle invasion by ECoV particles and immune-mediated muscle injury, presenting as myositis, although satisfactory proof of the direct invasion of SARS-CoV-2 into muscle cells is still lacking [11]. Lastly, muscle deconditioning due to immobility and corticosteroid treatment has been associated with diffuse atrophy at muscle biopsy [12]. A retrospective case study from China revealed that 10.7% of patients showed skeletal muscle injury during this acute phase as well as various neurological manifestations (36.4%) [13].

As a result, studies have mainly focused on the importance of nutritional intervention during the acute phase of COVID-19 in order to prevent clinical deterioration (review in [14]). However, the consequences of unintentional weight loss and resulting sarcopenia can also have major long-term functional impacts [15]. A large number of patients are being discharged from hospital following COVID-19 without a systematic assessment of their recovery and need for rehabilitation or further investigation to detect complications, in particular functional complaints related to loss of muscle strength, persistence of malnutrition and fatigue [16]. Medical teams have become increasingly aware of the importance of multidisciplinary care, taking into account fatigue and functional disability, but nevertheless overlooking the nutritional aspect and the importance of the decrease in muscle strength [1]. Nowadays, muscle weakness and fatigue appear as a frequent complaint among these patients [4,5].

In clinical practice, impairment of muscle strength is difficult to assess since, unlike weight, there is no objective assessment of muscle strength prior to disease. Such objective evaluation can be quite straightforward, i.e., using the grip test or chair lifting test. However, these tests are not easily implemented in the context of severe infection or in the patient’s home, and do not establish whether the alteration is linked to the disease or whether it existed beforehand. In view of the latter, we found it useful to use a self-evaluation muscle strength scale, asking the patient how he/she rated his/her muscle strength compared to before the disease. In a preliminary study, we found a concordance between the outcome of this subjective functional evaluation (Self Evaluation of Strength (SES)) and that of the grip test [17].

During the first wave of the pandemic, we rapidly established a post-COVID follow-up service collecting data to identify unmet health needs and to identify those requiring additional rehabilitation and/or investigation for complications. All of these patients received dietary counseling and adapted physical activity guidance, some of whom with severe disability and being re-admitted to post-care facilities. The others received physiotherapy assistance and nutritional supplementation.

The aim of this study was to evaluate, at 6 months post-discharge, the evolution of health status in the group of patients who had a persistent impairment at day 30: namely, patients with persistent weight loss (>5%) or impaired muscle strength (SES < 7/10), or with a Performance Status (PS) > 2. The objective was to analyze the frequency and nature of their symptoms persisting since the initial infection, their nutritional status as well as the evolution of their muscle strength and performance status.

## 2. Methods

### 2.1. Study Design and Participants

This study, conducted as part of a prospective cohort study, included all adult inpatients (≥18 years old) who were diagnosed with COVID-19 and admitted to an ICU or non-ICU unit for COVID-19 patients at the Nancy Brabois University Hospital between 1 March 2020 and 29 April 2020 and subsequently discharged alive from hospital. The diagnosis of COVID-19 was based on a positive SARS-CoV-2 RT-PCR test on a nasopharyngeal sample and/or on a typical chest CT scan [18]. The study was approved by the Research Commission of the University Hospital of Nancy and the requirement for informed consent was waived by the ethics commission. The ClinicalTrials.gov (accessed on 16 August 2021) identifier is NCT04451694.

To manage the flow of incoming patients, a certain number were transferred to other hospitals after a few days. At discharge from the intensive or acute care unit, some patients were admitted to post-acute care facilities for ongoing skilled nursing care and rehabilitation.

All hospital-discharged patients were offered a teleconsultation 30 days after discharge (D30) to assess their nutritional status and muscle function, the degree of disability linked to the degree of malnutrition and subjective functional loss, as well as limitation of daily activity estimated by the WHO performance status score. Consequently, patients with persistent weight loss (>5%) at D30 or with impaired muscle strength (Self Evaluation of Strength < 7/10) or a performance status (PS) ≥ 2 (group of impairment patients) were invited to be evaluated 6 months after discharge (by teleconsultation or in person).

### 2.2. Demographics, Comorbidities and Hospitalization Data Collection

Patient characteristics and hospitalization data were collected by manual review of electronic medical records. Epidemiological, demographic, laboratory and outcome data were extracted from electronic medical records during hospitalization. Sociodemographic data included age, sex, living alone or with others, occupational activity (active vs. unemployed and retired), smoking status (active or not) and daily alcohol consumption. Health characteristics included comorbidities (hypertension, diabetes mellitus, cerebrovascular disease, cardiovascular disease, chronic lung disease) as well as COVID-19 symptoms (anosmia and dysgeusia, diarrhea, dyspnea, asthenia, food aversion).

Hospitalization characteristics included: ICU admission (yes/no), time between symptom onset and hospitalization (days) and length of stay (days).

### 2.3. Recorded Symptoms

Asthenia at discharge, at day 30 and at 6 months post-discharge was evaluated using a fatigue visual analogue scale (VAS) (0–10) [19].

Dyspnea at discharge, at day 30 and at 6 months post-discharge was evaluated by the French adaptation of the American Thoracic Society Scale, according to 5 levels (from 1 to 5) [20].

Health status at day 30 and at 6 months was assessed using the WHO/Zubrod Performance Status Scale which rates patients from 0 to 4 [21].

Neurological symptoms were also explored (neuropathy, headache, impaired memory and concentration or cognitive impairment). 

Depression, anxiety and PTSD diagnoses were based on DSM-V criteria [22,23].

### 2.4. Nutritional Assessment

Nutritional status prior to hospitalization, on admission, at discharge, at day 30 and at 6 months post-discharge was assessed using anthropometric measurements (BMI: body mass index = body weight/height^2^) and weight loss (%) compared to weight prior to illness. In hospital, patients were weighed and their height was measured. On day 30 and at 6 months post-discharge, patients were instructed to use their own weighing scales.

### 2.5. SEFI and Self-Assessment of Muscle Function (SES) at Discharge, at Day 30 and at 6 Months Post-Discharge

Food intake was assessed using the 10-point verbal (AVeS) or visual (AViS) analogue scales (self-evaluation of food intake (SEFI)) [24], graded from 0 to 10. As suggested by Bouette et al. [24], an SEFI < 7 was considered as the cut-off value.

Self-assessment of strength (SES) was assessed at discharge, at day 30 and at 6 months post-discharge using the 10-point verbal (AVeS) or visual (AViS) analogue scales via teleconsultation for evaluating arm and leg strength in comparison to patient strength prior to hospitalization. In practice, patients were asked to evaluate their arm and leg strength in comparison with their strength prior to COVID-19. As suggested by Krznaric et al. [14], patients were asked about their degree of difficulty in lifting or carrying a weight, walking across the room, rising from a chair or bed, and to evaluate these difficulties using a 10-point verbal or visual analogue scale (10 = same strength as before illness and 0 = total loss of strength).

Assessment of physical activity was carried out by completing the International Physical Activity Questionnaire-Short Form (IPAQ-SF) [25] in order to estimate activity prior to COVID-19, with activity classified as low, moderate and high physical activity [25].

### 2.6. Malnutrition Diagnosis

Malnutrition diagnosis was made according to the GLIM criteria and French recommendations (at least 1 phenotypic criterion and 1 etiological criterion) [26,27]. Severe malnutrition was defined following the French recommendations as weight loss > 10% of weight before COVID-19 infection or a BMI < 17 (<18.5 for patients > 70 years old). Moderate malnutrition was defined as weight loss > 5% of weight before COVID-19 or BMI < 18.5 (<21 for patients > 70 years old) [27,28].

### 2.7. Statistical Analysis

Continuous variables are expressed as mean ± SD and categorical variables as absolute values and percentages. A paired Student’s *t*-test, chi-square (χ2), ANOVA, Fisher’s exact test, and Wilcoxon tests were used to compare the values of variables between groups as appropriate. Pearson’s chi-square test or Fisher’s exact test was used to assess the association between each of the discrete variables and the impairment status. *p* < 0.05 was considered statistically significant. Data were recorded on Excel files. Statistical analysis was performed using Statistical Analysis Software 9.4, SAS Institute Inc, Cary, NC, USA.

## 3. Results

Population description (Table 1 and Figure 1).

Of the 549 consecutive patients hospitalized for COVID-19 between 1 March and 29 April 2020, 23.7% died and 288 patients were at home at D30 post-discharge (Figure 1).

The mean age of the 288 interviewed patients was 59.8 ± 16.6 years, 54.2% of whom were male. The majority had a low physical activity assessed by IPAQ-SF prior to contracting COVID-19 (56.3%). Seventy-six percent of these patients were overweight or obese. According to the GLIM criteria, 20.7% were already undernourished on admission. The mean delay between the first symptom and admission was 9.7 ± 8.6 days. The most frequent comorbidities were cardiovascular diseases (hypertension, dyslipidemia, diabetes mellitus and coronary artery disease). At admission, anosmia was present in 42.7% of patients, dyspnea in 22.6%, cough in 25.0% and diarrhea in 9.3% of patients.

As reported previously [17], 13.2% of the patients hospitalized for a severe form of COVID-19 were still severely malnourished 30 days after hospital discharge (weight loss > 10% and/or BMI < 17). The highest predictive factors of persistent malnutrition were ICU stay and male sex. Moreover, 26.3% of these patients complained of impaired muscle strength and had a subjective functional loss evaluated at <7/10 using a 10-point verbal (AVeS) or visual (AViS) analogue scale. Lastly, 8.3% (n = 24) concomitantly exhibited a performance status (PS) ≥ 2, severe malnutrition and subjective functional loss.

At D30, 136 patients (47.2%) presented an impairment with persistent malnutrition or impaired muscle strength (SES < 7/10) or severe disability with a PS ≥ 2. Of the latter, 119 (91.0%) accepted phone or teleconsultations or in-person interviews at 6 months. Two patients died during this period, three were still hospitalized, eight were unreachable and there was one refusal (Figure 1).

The evolution of the characteristics of the 119 patients with impairment at D30 is shown in Table 2.

As shown in Table 2, patients with impairment at D30 regained an average of 3.6 kg body weight between D30 and 6 months, with mean weight and BMI returning to near baseline values (admission); on average, patients remained 1.4 kg lighter than on admission. Forty-three patients (36.0% of the 119 patients with impairment at D30) displayed persistent malnutrition at 6 months, with 18 patients exhibiting severe malnutrition (15.1%). On average, seventeen patients (14.3% of the 119 patients with impairment at D30) complained of a significant decrease in muscle strength (SES < 7/10 with a mean ± SD = 5.0 ± 2.6 for arm and 4.9 ± 2.6 for legs). Seventeen patients also had a performance status ≥ 2 (14.9% of the 119 patients with impairment at D30). SES increased between D30 and 6 months post-discharge, nearing the index value prior to COVID infection.

### 3.1. Long COVID Symptoms at 6 Months Post-Discharge

The most frequent symptoms were asthenia (16.0% had fatigue score > 5/10 at 6 months) and psychiatric disorders; 12 patients (10.0%) experienced mood disorders, anxiety, or post-traumatic stress syndrome. Nine patients (7.6%) presented prolonged pneumological symptoms (dyspnea), and five patients (4.2%) had neurological symptoms (neuropathy, headache, impaired memory and concentration or cognitive impairment).

### 3.2. Characteristics of Patients with Impairment at 6 Months Post-Discharge, Comparison with Recovered Patients

Overall, at 6 months, 53 patients presented a persistent impairment, as assessed by malnutrition and/or SES < 7 or performance status ≥ 2. The mean characteristics of this population with persistent impairment are described in Table 3 and were compared with the recovered patients (at discharge or at D30).

Comorbidity prevalence did not significantly differ between these two groups, except for obesity which was more frequent in the group with impairment at 6 months post-discharge (52.8% vs. 31.0%; *p* = 0.007), with these patients being admitted more frequently to ICUs (50.9% vs. 31.3%; *p* = 0.010). These patients had more difficulties eating after discharge (SEFI: 6.1 ± 3.0 vs. 7.4 ± 2.9 *p* = 0.0032) and exhibited greater weight loss, with an increased prevalence of malnutrition. SES scores were lower at discharge in this group (4.3 ± 2.4 vs. 5.3 ± 2.6 for arms; *p* = 0.013, and 4.0 ± 2.0 vs. 5.2 ± 2.7 for legs; *p* = 0.0045) with no other significant difference being observed.

## 4. Discussion

This is the first report on a prospective observational cohort study of COVID-19 specifically exploring nutritional status, subjective functional loss and disability at 6 months after discharge. We showed that, at D30, 138/288 of these patients (48%) presented persistent malnutrition (33%), subjective functional loss (26.3%) and/or performance status ≥ 2 (24.3%). At 6 months, 15% of the initial cohort remained malnourished despite nutritional counseling during hospitalization and ensuing dietary guidance, oral nutritional supplements, or relocation to rehabilitation centers, 6% complained of a significant decrease in muscle strength and 6% had a performance status > 2, or 18.5% of this cohort.

Contrary to our results, a study conducted in a cohort of severe COVID-19 patients assessing nutritional status 3 months after discharge from hospital [29] reported that only 8.4% of patients were malnourished at discharge and none at 3 months. Malnutrition was correlated with severity of the disease, indirectly inferable from an assessment of length of hospital stay and need for admission to ICU. However, in this latter study, malnutrition diagnosis was based on the 2015 ESPEN consensus [30], including different criteria for malnutrition than those of the international consensus (GLIM criteria [26]), in particular the postulation that patients with a BMI above 22 are not malnourished. There are several arguments against this latter assertion. The first is that patients suffering from obesity have, because of metabolic modifications (insulin resistance, fatty acid metabolism, hyperglycemia, etc.), higher protein catabolism than subjects of normal weight in the event of traumatism [31], which was also reported in acute COVID patients with obesity [32]. Secondly, accelerated muscle loss is a major factor of morbidity and mortality in obese COVID-19 patients [33]. On the contrary, in our study, obese subjects had a higher risk of functional loss or undernutrition 6 months after a severe COVID infection, as well as patients with a stay in intensive care.

In our cohort, 20.7% of patients were already undernourished at admission. Malnutrition tended to have begun during the initial phases of the disease occurring at home since, upon admission, patients declared significant involuntary weight loss when compared to their habitual weight [29]. However, it is not excluded that malnutrition may precede the infection. A recent study showed that patients with a recent history of malnutrition could be at higher risk of severe COVID-19 [34]. The majority of our patients had comorbidities that could lead to malnutrition. However, we did not observe any influence of these comorbidities on the evolution of their nutritional status. Weight loss and undernutrition were seemingly associated very early with a major decrease in food intake, which remained strongly impaired in half of the patients at hospital discharge, despite nutritional support. Caccialanza et al. [35] also showed that this reduced self-reported food intake prior to hospitalization and/or expected by physicians in the days after admission was associated with negative clinical outcomes in non-critically ill, hospitalized COVID-19 patients. The other component is the increased energy expenditure secondary to a major inflammatory syndrome that leads to hypercatabolism which, associated with reduced food intake and immobilization, significantly contributes to muscle atrophy and sarcopenia.

Acute sarcopenia may mostly affect patient prognosis and incur post-COVID-19 functional and physical deterioration. The degree of muscle mass and functional loss can be influenced by a multiplicity of factors, including the patient’s general pre-infection medical and functional condition, especially in older adults [10]. However, this functional condition prior to infection is typically not analyzed and/or is unknown. The subjective assessment of muscle strength by an analogue or numeric scale could therefore be useful to follow the evolution of muscle strength in these patients. Some patients still appeared very weak 6 months after the infection, despite advice to increase physical activity with protein support, or referral to physiotherapists or adapted physical activity therapists. Most of our patients had low physical activity assessed by IPAQ-SF prior to contracting COVID-19 (56.3%), which may represent a risk factor for severe COVID-infection [36,37].

Fifteen percent of patients presented neuro-psychiatric symptoms requiring treatment. These disorders are well-described in the literature and could be linked to a direct action of the virus on the nervous system [38,39].

Among the strengths and weaknesses of this analysis, the subjective evaluation of muscle strength is subject to debate. The strong point of such evaluation is to assess muscle function in the long term and to judge the evolution of muscle strength, without knowledge of the level of strength prior to the disease. We evaluated this tool in a small cohort of hospitalized patients [17] and while we were unable to identify the actual threshold of functional impairment, our assessment was nevertheless based on the threshold determined according to the questionnaire of Krznaric et al. [14] and on other subjective visual or numeric scales [40]. This threshold must, however, be validated in a large cohort study, notably in comparison with the evolution of muscle strength objectively measured by dynamometry.

Weighing patients at home is also a weak point due to the lack of weight scale control. Nevertheless, this measurement error was, *a priori*, the same for the patient’s usual weight prior to infection.

Lastly, we hypothesized that patients who were not impaired at discharge or who improved their nutritional status and muscle strength between discharge and D30 had little risk of having a worse outcome thereafter and were thus not contacted after 6 months. However, three additional deaths were observed during this latter period, albeit all linked to an underlying disease.

In conclusion, undernutrition and loss of muscle strength are symptoms of long COVID and should be considered in the clinical assessment of these patients. Although there are no current specific treatments for use in patients who have been hospitalized for COVID-19, treatments should focus on nutritional support and rehabilitation exercises whenever possible to prevent long-term disability as a result of acute illness due to COVID-19, as well as on the management of sarcopenia. The data described herein may assist in the identification of patients outside of expected recovery trajectories who could benefit from additional rehabilitation and/or further investigation to detect post-COVID nutritional complications. Of these outlying patients, obese subjects appear particularly at risk, as well as patients who have stayed in intensive care.

## Figures and Tables

**Figure 1 nutrients-13-03964-f001:**
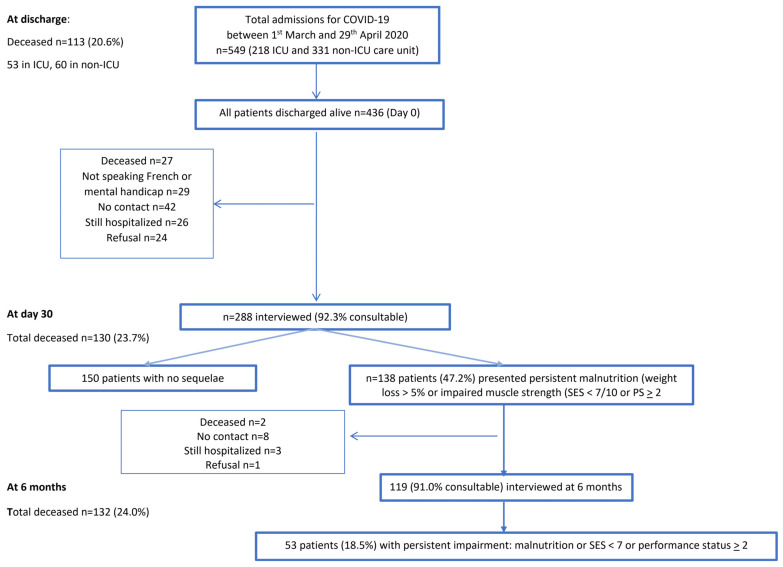
Flowchart of patients included in the study.

**Table 1 nutrients-13-03964-t001:** Characteristics of the whole population (n = 288).

Variables	n and %
Age (years)	59.8 ± 16.6
Sex (F/M)	132/156 (45.8%/54.2%)
Living alone	54 (18.8%)
Couple	132 (45.8%)
Family	96 (33.3%)
Retirement home	6 (2.1%)
BMI Class	
18.5–24.9	69 (24.3%)
25–29.9	112 (39.4%)
>30	103 (36.3%)
IPAQ (prior to COVID-19)	
Low	162 (56.3%)
Moderate	82 (28.5%)
High	44 (15.3%)
ICU	102 (35.4%)
Active smoking	12 (4.2%)
Daily alcohol	14 (4.9%)
Comorbidities	
HBP	113 (39.2%)
Coronary heart disease	41 (14.2%)
Dyslipidemia	64 (22.2%)
Diabetes mellitus	48 (16.7%)
Renal failure	23 (8.0%)
Stroke	9 (3.1%)
Asthma	14 (4.9%)
Apnea	19 (6.6%)
Chronic obstructive bronchitis	13 (4.5%)
Respiratory failure	10 (3.5%)
Active cancer	30 (10.4%)
Neurological disease	13 (4.5%)

BMI, body mass index; IPAQ, International Physical Activity Questionnaire; ICU, intensive care unit; HBP, high blood pressure.

**Table 2 nutrients-13-03964-t002:** Characteristics and evolution of nutritional status at 6 months in the sub-group of patients with impairment at day 30 after severe COVID-19 (n and %).

	Admission (n = 119)	At Discharge (n = 119)	D30 Home (n = 119)	At 6 Months (n = 119)	*p*
Weight (kg)	82.6 ± 19.06	76.5 ± 16.1	77.6 ± 15.9	81.2 ± 17.9	<10^−3^
BMI	28.7 ± 5.9	26.7 ± 5.0	27.4 ± 5.0	28.3 ± 5.5	<10^−3^
Weight variation (%)	−3.7 ± 4.9	−5.1 ± 5.9	−7.4 ± 5.0	−3.6 ± 5.9	<10^−3^
Malnutrition					<10^−3^
No	68/119 (57.1%)	23/119 (19.3%)	22/119 (18.5%)	76/119 (63.9%)	
Moderate	21/119 (17.6%)	42/119 (35.3%)	43/119 (36.1%)	25/119 (21.0%)	
Severe	12/119 (10.1%)	54/119 (45.4%)	54/119 (45.4%)	18/119 (15.1%)	
SEFI		7.2 ± 3.0	9.6 ± 1.2	9.8 ± 0.9	<10^−3^
SES hands		4.2 ± 2.4	6.9 ± 2.1	9.1 ± 1.7	<10^−3^
SES legs		4.0 ± 2.3	6.8 ± 2.1	9.1 ± 1.6	<10^−3^
Subjective functional loss (SES < 7)%		79.8%	55.5%	14.3%	<10^−3^
Asthenia (VAE)> 5/10 (%)			3.2 ± 3.0 35/119 (29.4%)	1.7 ± 2.5 19/119 (16.0%)	<10^−3^ 0.020
Performance Status					<10^−3^
0		22 (18.5%)	30 (25.2%)	70/119 (58.8%)	
1		44 (37.0%)	44 (37.0%)	27/119 (22.7%)	
2		29 (24.4%)	29 (24.4%)	9/119 (7.6%)	
3–4		24 (20.2%)	16 (13.4%)	8/119 (6.7%)	

D30, day 30; BMI, body mass index; SEFI, self-evaluation of food intake; SES, self-evaluation of strength.

**Table 3 nutrients-13-03964-t003:** Comparison of the characteristics of recovered patients and of the group of patients with impairment at 6 months.

	Recovered Patients	Patients with Impairments at 6 Months	
Variables	N = 217 (80.4%)	N = 53 (19.6%)	*p*
Symptoms at discharge			
Dyspnea ≥ 2	21 (9.6%)	3 (5.7%)	NS
Anosmia at discharge	95 (43.8%)	18 (34%)	NS
Diarrhea	10 (4.6%)	2 (3.8%)	NS
cough	16 (7.4%)	4 (7.5%)	NS
Obesity (BMI ≥ 30) (%) at admission	31.0%	52.8%	0.007
ICU admission	31.3%	50.9%	0.010
SEFI at discharge	7.4 ± 2.9	6.1 ± 3.0	0.003
Dietary aversion at discharge	19 (8.8%)	8 (15.1%)	NS
Weight loss (%)			
Before admission	−2.8% ± 4.1%	−5.5 ± 7.0	0.001
Total at discharge	−5.2% ± 4.7%	−10.5 ± 6%	<0.001
Malnutrition at discharge (%)	102 (49.3%)	43 (82.6%)	0.001
Moderate	62 (30.0%)	15 (28.8%)	
Severe	40 (19.3%)	28 (53.8%)	
SES at discharge			
Arms	5.3 ± 2.6	4.3 ± 2.4	0.013
Legs	5.2 ± 2.7	4.0 ± 2.0	0.004
SES < 7	138 (71.9%)	46 (92%)	0.003

Fisher’s exact test for qualitative variables, Student’s *t* test for quantitative variables. SEFI, self-evaluation of food intake; SES, self-evaluation of strength.

## Data Availability

The data that support the findings of this study are available from the corresponding author upon reasonable request.

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
