# Peer review of "Long-Term Evolution of Malnutrition and Loss of Muscle Strength after COVID-19: A Major and Neglected Component of Long COVID-19"

_nutrients, 2021, doi:10.3390/nu13113964_

Round 1
Reviewer 1 Report
This is an interesting study, but the paper needs significant improvement before being published. I recommend the authors refer to the STROBE reporting guidelines to ensure your paper meets reporting requirements. Please find my comments below:
ABSTRACT
- First sentence - Reword to: “Post-acute consequences of COVID-19, also termed long COVID, include signs and symptoms persisting for more than 12 weeks with prolonged multisystem involvement; but most often, malnutrition is ignored.”
- Use ‘at month six’ instead of ‘at M6’
- Use of decimal places is inconsistent; report all to 1 decimal place (i.e. 91.0%)
- Reword this sentence to: “Obesity was more frequent in patients with disabilities than those without (52.8% vs.. etc.)”
- Reword this sentence to: “Among those with persistent symptoms, 10% had….”
- There is data in the abstract that is not presented in the Results section of the paper; i.e. information on the 549 patients, 23.7% who died, number of patients who were home. You also report on nutrition interventions / nutrition care received in the abstract but not in the main results. Please add all this information to the results section of the paper.
INTRODUCTION
- Overall, the introduction needs more detail. Can you describe findings from other research related to long-term nutrition-related effects of COVID-19 (not just your own data)? And state why monitoring nutritional/functional outcomes is important / significant (e.g. to patients, health services, economy)? You need further background and rationale for the study here.
- Sentence 2 – reword to: “The most commonly described…”
- Last paragraph – reword to: “All of these patients received dietary counseling and adapted physical activity guidance, some of whom with severe disability were re-admitted to post-care facilities.”
- Aim is not stated clearly – please describe.
- I would reconsider the use of the term ‘disability’ as this may be interpreted as the more traditional use of the word. You could use the word ‘impairment’ instead. Or, if you must use the term disability, please define it early on in the introduction (i.e. what you mean by ‘disability’).
METHOD
- Methods are unstructured and therefore a little hard to follow. Please consider using subheadings, including Study design; Setting; Participants; Outcomes; Data collection (and you already have Statistical analysis heading).
- Methods are not adequately described. Please use the STROBE reporting guidelines to ensure you have included all relevant information. For example:
- Study design
- Study setting, location, dates
- Participants (including eligibility criteria, sources and methods of selection and recruitment), how sample size was determined
- Data collection – describe how data were collected from participants (in more detail than currently provided)
- What criteria were used to diagnose malnutrition?
- How was height measured?
- How were all the surveys administered?
RESULTS
- Results are missing key information. Please use STROBE guidelines to guide reporting. Results also need to be structured better.
- There is no description of participants or their characteristics. How many patients were included in the study? How many completed the study (i.e. had all data collected at 6 month follow up)? What was their demographics (age, gender, hospital length of stay, ICU admission, co-morbidities etc.)?
- Please describe data availability (i.e. how much missing data at each time point)
- First sentence – reword to: “Of 288 patients, 136 (47.2%) presented with malnutrition, impaired muscle strength, or severe disability at D30; 119 (91%) of whom accepted phone or teleconsultations or in-person interviews at six months. Two patients died during this period and three were still hospitalized.
- Second sentence – reword to: “The most frequent symptoms were psychiatric; 12 patients (10%) experienced mood disorders, anxiety or post-traumatic stress syndrome. Nine patients (7.6%) presented with prolonged pneumological symptoms (dyspnea), and five patients (4.2%) had neurological symptoms (neuropathy, headache, impaired memory and concentration or cognitive impairment).
- Third sentence – reword to: “As shown in Table 1, patients regained an average of 3.6 kg body weight between D30 and M6, with mean weight and BMI almost returning to baseline (admission); on average patients remained 1.4 kg lighter than on admission.”
- Please use consistent terms to describe COVID-19 (i.e. change SARS-COVID-19 in Table 1 to COVID-19)
- For the sentence: “Forty-three patients (36.0% of this group and 14.9% of the whole population) had persistent malnutrition, within 18 patients exhibiting severe malnutrition (15.1 % of this group and 6.2 % of the whole population)” – it is unclear which group you are talking about when you say “this group” – is this the overweight patients? Those with disability? Please be clear.
- Top of page 4: Again, it is unclear which group you are talking about here. Please clarify. Also, I would start with the findings on loss of strength at D30, and present the M6 data afterwards, so results are presented in logical/consecutive fashion.
- Table 2 should be repeated for ALL participants and be presented at the beginning of the Results section
- Page 5 – for the sentence: “Obesity was more frequent in this group…” – please state which group you are referring to.
- The results of the nutrition-related questions outlined in the Methods are not presented
- Food intake data are not presented
- Unclear how ‘dietary aversion’ was calculated (this should be reported in Methods)
DISCUSSION
- Discussion needs more work. There is no reference to / discussion of previous literature. You could find studies to compare and contrast to your findings.
- I would start the discussion with the finding that nutritional status and muscle strength are significantly poorer at 30D than baseline after COVID-19 infection. Then state that this improved at 6 months for most individuals.
- Reword to: “…nevertheless presented with persistent malnutrition…”
- Limitations are not adequately described. With the lack of detail in the methods, it’s hard for me to discern all limitations, but some examples that I can see include: having patients measure their weight on scales at home (not standardised), having patients do a self-assessment of strength and food intake, and most data is self-report and therefore subjective. Please state that the results of this study should be interpreted with the limitations in mind.
Reviewer 2 Report
I congratulate the authors on the work done on this relevant subject. However, I believe the paper needs to be revised, The topic is interesting, although not novel, and the manuscript is not well written. I would like to contribute with some comments, suggestions and questions to the authors:
- Tables: I suggest the authors standardize the number of decimals in p values in all tables and keep 3 decimals.
- Statistical analysis "Variables significant at a 0.05 level were subsequently used in multivariate analyses" - where is this model?
- Discussion: I would like the authors to interpret and comment result in the discussion. Where is References? In discussion you used only one reference. The discussion should be complete rewrite. In the discussion section you only repeat result.
Author Response
Please see the attchment

Round 2
Reviewer 1 Report
Paper is much improved and is now acceptable for publication.
Reviewer 2 Report
I accepted in present from. Without comments. Author's made adjustment according to my major comments.